# Targeting Adaptive IRE1α Signaling and PLK2 in Multiple Myeloma: Possible Anti-Tumor Mechanisms of KIRA8 and Nilotinib

**DOI:** 10.3390/ijms21176314

**Published:** 2020-08-31

**Authors:** Yusuke Yamashita, Shuhei Morita, Hiroki Hosoi, Hiroshi Kobata, Shohei Kishimoto, Tatsuya Ishibashi, Hiroyuki Mishima, Akira Kinoshita, Bradley J. Backes, Koh-Ichiro Yoshiura, Feroz R. Papa, Takashi Sonoki, Shinobu Tamura

**Affiliations:** 1Department of Hematology/Oncology, Wakayama Medical University, Wakayama 641-8509, Japan; yyyamash@wakayama-med.ac.jp (Y.Y.); h-hosoi@wakayama-med.ac.jp (H.H.); ketunai@wakayama-med.ac.jp (H.K.); sonoki@wakayama-med.ac.jp (T.S.); 2First Department of Internal Medicine, Wakayama Medical University, Wakayama 641-8509, Japan; kishimot@wakayama-med.ac.jp (S.K.); t1484@wakayama-med.ac.jp (T.I.); 3Department of Human Genetics, Atomic Bomb Disease Institute, Nagasaki University, Nagasaki 852-8523, Japan; hmishima@nagasaki-u.ac.jp (H.M.); akino@nagasaki-u.ac.jp (A.K.); kyoshi@nagasaki-u.ac.jp (K.-I.Y.); 4Department of Medicine, University of California, San Francisco, CA 94158, USA; Bradley.Backes@ucsf.edu (B.J.B.); Feroz.Papa@ucsf.edu (F.R.P.); 5Diabetes Center, University of California, San Francisco, CA 94158, USA; 6Quantitative Biosciences Institute, University of California, San Francisco, CA 94158, USA

**Keywords:** multiple myeloma, unfolded protein response, IRE1α, KIRA8, nilotinib, PLK2

## Abstract

Background: Inositol-requiring enzyme 1α (IRE1α), along with protein kinase R-like endoplasmic reticulum kinase (PERK), is a principal regulator of the unfolded protein response (UPR). Recently, the ‘mono’-specific IRE1α inhibitor, kinase-inhibiting RNase attenuator 6 (KIRA6), demonstrated a promising effect against multiple myeloma (MM). Side-stepping the clinical translation, a detailed UPR phenotype in patients with MM and the mechanisms of how KIRA8 works in MM remains unclear. Methods: We characterized UPR phenotypes in the bone marrow of patients with newly diagnosed MM. Then, in human MM cells we analyzed the possible anti-tumor mechanisms of KIRA8 and a Food and Drug Administration (FDA)-approved drug, nilotinib, which we recently identified as having a strong inhibitory effect against IRE1α activity. Finally, we performed an RNA-sequence analysis to detect key IRE1α-related molecules against MM. Results: We illustrated the dominant induction of adaptive UPR markers under IRE1α over the PERK pathway in patients with MM. In human MM cells, KIRA8 decreased cell viability and induced apoptosis, along with the induction of C/EBP homologous protein (CHOP); its combination with bortezomib exhibited more anti-myeloma effects than KIRA8 alone. Nilotinib exerted a similar effect compared with KIRA8. RNA-sequencing identified *Polo-like kinase 2* (*PLK2*) as a KIRA8-suppressed gene. Specifically, the IRE1α overexpression induced PLK2 expression, which was decreased by KIRA8. KIRA8 and PLK2 inhibition exerted anti-myeloma effects with apoptosis induction and the regulation of cell proliferation. Finally, PLK2 was pathologically confirmed to be highly expressed in patients with MM. Conclusion: Dominant activation of adaptive IRE1α was established in patients with MM. Both KIRA8 and nilotinib exhibited anti-myeloma effects, which were enhanced by bortezomib. Adaptive IRE1α signaling and PLK2 could be potential therapeutic targets and biomarkers in MM.

## 1. Introduction

Despite considerable improvements attained in multiple myeloma (MM) treatment due to the development of novel agents, most patients still relapse and become refractory to the therapy [1]. As MM cells produce extensive monoclonal immunoglobulin (Ig) proteins, they are considered to possess a well-developed endoplasmic reticulum (ER) capable of enduring excessive stress. The accumulation of unfolded and misfolded proteins in the ER stimulates an intracellular signaling pathway, known as the unfolded protein response (UPR), resulting in transcriptional and translational regulations of ER-associated proteins [2,3,4]. Inositol-requiring enzyme 1α (IRE1α) is an evolutionally conserved UPR sensor protein, which is highly activated in myeloma cells [5,6,7]. Under remediable ER stress, IRE1α plays a vital role in preserving cell viability and functioning by triggering a frameshift splicing of the X-box Binding Protein 1 (*XBP1*) mRNA. The translated spliced form *XBP1* (*sXBP1*) transcriptionally upregulates genes to manage protein folding demand and sustain the cell, known as adaptive UPR (A-UPR) [8,9,10].

We developed an allosteric IRE1α kinase inhibitor, kinase-inhibiting RNase attenuator 6 (KIRA6), which attenuates terminal UPR (T-UPR) to preserve cell function and viability under irremediable ER stress, which ameliorated the disease state in both diabetic and retinitis pigmentosa model animals [8]. We recently revealed that ABL kinases localize to the ER membrane and scaffold IRE1α to activate the UPR under ER stress [10]. To confirm this, Imatinib, a Food and Drug Administration (FDA)–approved tyrosine kinase inhibitor (TKI) targeting BCR-ABL optimized for chronic myelogenous leukemia (CML), exhibited an inhibitory effect against IRE1α activity [10]. Nilotinib, an equally selective but more potent inhibitor than imatinib, also exhibited an inhibitory effect on *XBP1* mRNA splicing, T-UPR, and apoptosis at lower concentrations than imatinib [10]. Subsequently, we illustrated that both a mono-selective inhibitor of IRE1α, KIRA8 (also known as compound 18) [11], and imatinib could inhibit the ABL–IRE1α axis to preserve β-cells from T-UPR and reverse autoimmune diabetes [10]. However, the anti-myeloma effects of KIRA8 and these TKIs remain only partially explored.

Although the potential of the IRE1α–sXBP1 pathway as a therapeutic target in several types of cancer has been extensively recognized [12], the detailed phenotypes of UPR in the patients with MM and the effects of IRE1α inhibitors on MM remain debatable [4,13]. Notably, a study recently demonstrated the anti-tumor activity of KIRA8 against MM in human myeloma cells, a xenograft model, and patient-derived CD138^+^ myeloma cells [14]. Despite promising effects, the mechanism underlying the work of KIRA8 against MM remains unclear. More importantly to clinical translation, determining whether FDA-approved TKIs work equally to other IRE1α inhibitors in myeloma cells is challenging. This study aimed to (i) characterize UPR activities in the bone marrow (BM) of patients with MM, and (ii) investigate the molecular mechanisms of how KIRA8 works and whether nilotinib exhibits anti-cancer effects in human myeloma cells.

## 2. Results

### 2.1. UPR Signaling in the BM of Patients with Newly Diagnosed Multiple Myeloma (NDMM)

As human myeloma cells adapt to chronic ER stress and continually activate IRE1α–XBP1 signaling [2,3,4], we determined which endogenous UPR signaling was induced in the BM of patients with newly diagnosed multiple myeloma (NDMM) compared with the control subjects (Appendix A). We retrospectively observed that the expression of *sXBP1* and the ER chaperone *BIP* mRNA (another A-UPR marker [10]) were upregulated in the BM of patients with NDMM, whereas they were sustained in the control subjects (Figure 1A,B). Generally, chronic and prolonged ER stress activated another sensor, protein kinase R-like endoplasmic reticulum kinase (PERK), to induce apoptosis through activating transcription factor 4 (ATF4) and C/EBP homologous protein (CHOP) [2,3,4]. In patients with NDMM, the elevation of the mRNA expression levels of *ATF4* and *CHOP* was limited (Figure 1C,D). Despite A-UPR induction, the mRNA expression levels of thioredoxin interacting protein (TXNIP), a T-UPR marker regulated by IRE1α and PERK, were not significantly increased (Figure 1E) [15,16]. These findings suggested that A-UPR in the IRE1α pathway is dominantly activated in the BM of patients with NDMM.

### 2.2. Effects of KIRA8 and PERK Inhibitors on Human Myeloma Cells

To determine the effects of KIRA8 on human myeloma cells, we used IM-9 cells, which exhibit splicing of *XBP1* mRNA even at the baseline (21.8% of spliced to total *XBP1* mRNA ratio) (Figure 2B). To induce global UPR, we used thapsigargin, which inhibits ER Ca^2+^-dependent ATPase. In IM-9 cells, KIRA8 inhibited *sXBP1* mRNA, both at the baseline and even under the robust UPR, with 500 nM of thapsigargin (Figure 2B). In IM-9 cells, 10 μM of KIRA8 markedly decreased the cell viability analyzed by the cell counting kit-8 (CCK-8) assay and trypan blue exclusion assay, along with the apoptosis induction (Figure 2C–E, Appendix A). 

Recently, we reported that KIRA8 led to the reciprocal hyperactivation of PERK and apoptotic signaling, halting tumor growth and survival during mild ER stress in pancreatic neuroendocrine tumors (p-NET) [17]. Surprisingly, KIRA8 increased the PERK-related *CHOP* mRNA expression in a dose-dependent manner in IM-9 cells, as shown in p-NET (Figure 2F). Notably, the IM-9 cells were mildly ER-stressed at the baseline (Figure 2B), suggesting that both the IRE1α and PERK pathways were continuously activated at the baseline. To investigate the compensatory signaling through a parallel pathway of IRE1α and PERK under mild ER stress, we further assessed the effect of two highly selective PERK inhibitors, GSK2606414 and AMG PERK 44 (AMG) [18,19]. 

Both GSK2606414 (at 10 μM) and AMG (at 25 μM) significantly decreased the cell viability in IM-9 cells (Figure 2G). At this point, AMG elevated the expression levels of *sXBP1* mRNA, suggesting compensation through the IRE1α pathway (Figure 2H). To further validate the effect of KIRA8 in other human myeloma cell lines, we used KMS-11, KMS-12-PE and KHM-11 cells. KIRA8 decreased the cell viability and increased apoptosis in all three cell lines (Figure 2I,J). Together, the KIRA8 and PERK inhibitors demonstrated anti-myeloma effects with induction of the compensational pathway.

### 2.3. Effects of Combined Treatment with KIRA8 and Bortezomib in Human Myeloma Cells

In human myeloma cells, bortezomib induces UPR signaling activated by the accumulation of misfolded proteins within the ER [2,3,4]. Bortezomib enhances the anti-myeloma effects of several chemotherapeutic agents. Thus, we analyzed the combined effect of KIRA8 and bortezomib in IM-9 cells. The combined treatment with bortezomib and KIRA8 at 10 μM decreased the cell viability more than KIRA8 alone (Figure 3A, Appendix A). Next, we determined whether the observed effects of the combined treatment on the cell viability could be related to apoptosis. The combined treatment markedly increased the apoptosis of IM-9 cells compared with bortezomib alone (Figure 3B,C). The effect on apoptosis was highlighted for the combination treatment with 5 nM of bortezomib and 10 μM of KIRA8 (Figure 3B,C). As previous studies demonstrated the UPR induction by bortezomib [2,3,4], we analyzed the effect of KIRA8 in combination with thapsigargin in IM-9 cells. Like bortezomib, the combination of KIRA8 and thapsigargin exhibited stronger effects than KIRA8 alone on the cell viability reduction and the apoptosis induction (Figure 3D, Appendix A). Thus, the combination of KIRA8 and bortezomib exhibited stronger anti-tumor effects on myeloma cells than KIRA8 alone.

### 2.4. Nilotinib Exhibits Anti-Myeloma Effects

Next, from the clinical aspect, we investigated whether nilotinib could induce anti-myeloma effects as KIRA8 does [10]. We selected nilotinib with these two reasons: First, among the FDA-approved drugs, high-throughput screening using a DELFIA assay identified nilotinib as a promising candidate inhibitor of IRE1α autophosphorylation [20]. Second, we previously demonstrated that nilotinib exerted robust inhibition of IRE1α activity by anchoring ABL in the cytosol with an IC50 of 3.21 μM for attenuating XBP1 splicing [10]. Surprisingly, nilotinib reduced cell viability with 10.37 μM of IC50 showing an induction of apoptosis (Figure 4A,B). Nilotinib was confirmed to inhibit *sXBP1* mRNA expression as we previously reported a strong inhibitory effect of IRE1α activity in pancreatic β-cells, whilst it induced *CHOP* mRNA expression (Figure 4C). Finally, the anti-myeloma effects of nilotinib were enhanced by bortezomib (Figure 4D). In summary, similarly to KIRA8, FDA-approved nilotinib exhibited novel anti-myeloma effects enhanced by bortezomib.

### 2.5. Gene Expression Profile Induced by KIRA8 in Human Myeloma Cells

To explore the mechanisms of anti-myeloma effects of KIRA8, the transcriptome profiling was examined by the RNA-sequence (RNA-seq) analysis in IM-9 cells. A heatmap visualized the expression patterns of the 64,985 identified differentially expressed genes (DEGs), which were then hierarchically categorized into 17 clusters through cluster analysis (Figure 5A, Appendix A). The DEGs in cluster 16 were those inhibited by all KIRA8-based treatments (Figure 5A, dotted square). We further identified the 30 most strongly downregulated protein-coding DEGs in IM-9 cells treated with KIRA8 alone (Figure 5B). Amongst the top 30 abundant genes, we focused on the *PLK2* in cluster 16, a member of the ‘polo’ family of serine/threonine protein kinases, which plays a vital role in normal cell division (Figure 5B, arrow) [21]. The 64,985 identified DEGs were classified into different functional categories based on the Gene Ontology (GO) term enrichment analysis for molecular function. The most significantly downregulated DEGs were associated with the GO terms negative regulation of (i) structural constituent of ribosome, (ii) protein kinase activity, and (iii) protein serine/threonine kinase activity (LogP < −5; Figure 5C, Appendix A). Among the genes associated with the kinase activity, *PLK2* was the most significantly downregulated by the KIRA8 treatment.

### 2.6. The Regulation of PLK2 by KIRA8 and the Effects of Its Inhibition on MM Cells

To confirm the results of RNA-seq analysis, we further examined the *PLK2* gene expression in IM-9 cells. KIRA8 suppressed the *PLK2* expression, both at the mRNA and protein levels (Figure 5D,E), and demonstrated a limited effect on the mRNA expression of *PLK1* (Figure 5D), the most intensively explored PLK family member [21,22]. Next, we used doxycycline (DOX)-inducible IRE1α-overexpressing INS-1 insulinoma cells to confirm the specific effect of IRE1α on PLKs [23]. Despite the limited effect on *PLK1* mRNA, the IRE1α overexpression significantly increased, both at the mRNA and protein levels of the *PLK2* expression (Figure 6A,B). Conversely, AMG increased the mRNA expression of *PLK2*, but not *PLK1*, potentially under the induction of *sXBP1* or the inhibition of regulated IRE1-dependent decay (RIDD) (Figure 2H, Figure 6C) [24].

The PLK family, including *PLK1* and *PLK2*, is involved in cell-cycle progression [21,22]. Accordingly, we examined these biological characteristics of PLK2 in myeloma cells. A PLK2-dominant inhibitor, TC-S7005, significantly reduced the cell viability and increased apoptosis in IM-9 myeloma cells (Figure 6D,E) [25]. In addition, TC-S7005 reduced the fraction of these cells in the G_2_/M phase (Figure 6F). Likewise, the upstream IRE1α pathway inhibition by KIRA8 decreased the cells in the G_2_/M phase, as shown in the PLK2 inhibition (Figure 6G). Together, *PLK2* expression, but not *PLK1*, was regulated by IRE1α, and its pharmacological inhibition induced apoptosis, demonstrating the cell-cycle arrest in human myeloma cells.

### 2.7. High Expression of PLK2 mRNA and Protein in NDMM

Finally, encouraged by the regulatory effect of IRE1α on PLK2, we assessed if the *PLK2* expression was, indeed, increased in patients with myeloma. As anticipated, the *PLK2* mRNA levels were significantly increased (>20 times) in the BM of patients with NDMM who attained a complete response receiving bortezomib-based treatment and high-dose melphalan compared with those of control subjects (Figure 6H). For further confirmation, we performed immunohistochemical analysis with the anti-PLK2 antibody in a BM clot of a patient with NDMM (Case MM-11). In this study, the PLK2 protein was highly expressed in BM cells, which were merged with CD138^+^ cells (Figure 6I). Overall, these findings suggested that the mRNA and protein expressions of *PLK2* were increased in human myeloma cells.

## 3. Discussion

Although the IRE1α–XBP1 pathway has been used as a therapeutic target, the expression of UPR markers remains only partially explored in patients with MM. This study demonstrated that A-UPR markers in the IRE1α pathway were dominantly increased over the PERK pathway in patients with NDMM compared with the control subjects [26,27]. Conversely, TXNIP, a T-UPR marker, was not increased. Motivated by these findings, we further explored the effect of the IRE1α–XBP1 pathway on the anti-tumor activity against MM by using the novel ‘mono’-specific inhibitor of IRE1α, KIRA8, in this study.

To date, several compounds targeting the IRE1α RNase activity have exhibited anti-tumor activity in in vitro and xenograft models [3,13,28,29,30]. Remarkably, KIRA8 was recently demonstrated to decrease the cell viability and growth of human myeloma cells and exhibit anti-tumor activity in a xenograft model [14]. Here, we confirmed the anti-myeloma effects of KIRA8 as in the previous report [14]. Next, we showed that specific inhibition of the IRE1α kinase domain by KIRA8 was coupled with CHOP expression, which could be associated with PERK activation, whereas selective PERK inhibitors conversely decreased the cell viability with the activation of IRE1α. Similar reciprocal effects of IRE1α–PERK were reported in p-NET [17]. Thus, in neoplasms driven from professional secretory cells, such as myeloma or β-cells, the balance of IRE1α–PERK activities could be essential for the survival and proliferation of tumors under basal ER stress.

Bortezomib was reported to exhibit an additional effect on growth suppression in myeloma cells in which the IRE1α kinase domain is knocked down [14]. Consistently, our study demonstrated that the combination of KIRA8 and bortezomib more strongly induced apoptosis in myeloma cells compared with monotherapy with either agent. In daily clinical practice, proteasome inhibitors are primarily used in combination with antibody drugs and other molecular-targeted drugs [31]. Hence, KIRA8 could be a novel molecular-targeted drug for MM, and its combination with proteasome inhibitors is considered a potent regimen.

Nilotinib is a close analogue of imatinib with approximately 20-fold higher potency inhibition of BCR-ABL, the oncogenic driver in CML [32]. Recently, we demonstrated that nilotinib attenuated IRE1α signaling under prolonged ER stress by anchoring ABL in the cytosol in islet β-cells [10]. In our study, the ABL–IRE1α signaling inhibition by nilotinib resulted in anti-myeloma effects, demonstrating the PERK pathway activation and enhanced cytotoxic effects with bortezomib. Although nilotinib has not been approved for MM treatment yet, its novel molecular mechanism could have the potential to expand the repertoire of therapeutic options in patients with MM.

The comprehensive transcriptome analysis by RNA-seq highlighted PLK2 as a key downstream target of the IRE1α kinase activation, possibly followed by a XBP1 or RIDD target [24]. Although there is a limitation in the cluster analysis of RNA-seq data that we performed the limited combination of KIRA8 based-treatment without bortezomib or thapsigargin alone, the PLK2 was a non-biased top-hit target by KIRA8 (Figure 5B) and its reduction by KIRA8 were clearly confirmed by quantitative PCR and western blot (Figure 5D,E). 

Our findings demonstrated that KIRA8 treatment suppressed the transition to the G_2_/M phase in myeloma cells, followed by the PLK2-dominant inhibitor exhibiting a similar effect as KIRA8. Compared with the abundant evidence of PLK1 inhibitors against several cancer types [22,33], the evidence regarding PLK2 inhibitors is limited. Based on the results of this and a limited number of preclinical studies, PLK2 could also serve as a potential therapeutic target for several cancer types by regulating apoptosis and cell proliferation [34].

This study has some limitations worth acknowledging. First, the sample size was small. Nevertheless, this is the first pilot study to demonstrate that the *PLK2* expression was increased in the BM of patients with myeloma, whereas it was constant in those of control subjects. In addition, a recent study reported that high *PLK2* expression correlated with poor outcomes in colorectal cancer (CRC) and served as a prognostic biomarker [34]. Our study raises the possibility that high *PLK2* expression could be a novel biomarker for not only CRC but also MM. Thus, further studies are warranted to establish the role of PLK2 as a biomarker and, more importantly, to determine if *PLK2* expression correlates with the outcomes in MM.

In conclusion, this study demonstrated the dominant activation of the adaptive IRE1α pathway in patients with NDMM and the anti-myeloma effects of KIRA8 and nilotinib with the reciprocal induction of CHOP. In addition, this study identified PLK2 as a novel key target of IRE1α, demonstrating its inhibitor-induced anti-myeloma effects. Overall, this study supports the promising therapeutic strategy with KIRA8 and nilotinib against MM by providing molecular and human pathological evidence regarding MM, as well as the likelihood of PLK2 as a novel therapeutic target and biomarker for MM.

## 4. Materials and Methods

### 4.1. Clinical Data

We retrospectively assessed 27 patients with NDMM at Wakayama Medical University (Wakayama, Japan) from January 2015 to January 2019. Among these patients with NDDM, we conducted a cross-sectional observational study of 11 consecutive patients (7 males and 4 females) receiving bortezomib- or lenalidomide-based induction treatment and having no exclusion criteria described below. We collected the patients’ pre-treatment BM samples. The exclusion criteria were; inaccessible or incomplete medical records, less than 10% myeloma cells in BM aspiration, and low quality of RNA obtained from the BM sample. The median age of MM onset was 63 (range: 43–77) years. All patients with MM were diagnosed according to the diagnostic criteria developed by the International Myeloma Working Group [35]. In addition, we enrolled six patients who were newly diagnosed with limited-stage diffuse large B-cell lymphoma, who had apparently normal BM trephine specimens, as the control group and collected their samples. The sample collection was approved by the hospital’s ethics committee (approval number: 57, approved on 31 July 2015), and we obtained written informed consent from all study participants. This study was conducted in accordance with the Declaration of Helsinki guidelines. Appendix A summarizes the patients’ characteristics.

### 4.2. Cell Line and Cell Culture

IM-9 cells (Cat# JCRB0024, RRID: CVCL_1305), human myeloma cells, were purchased from the Japanese Collection of Research Bioresources Cell Bank/National Institute of Health Sciences (Tokyo, Japan). In addition, KMS-11 (RRID: CVCL_2989), KMS-12-PE (RRID: CVCL_1333), and KHM-11 (RRID: CVCL_A633) cells were kindly provided by Drs. Hata and Kawano, the Department of Hematology, Kumamoto University (Kumamoto, Japan) [6]. The cell lines were maintained in RPMI-1640 (Sigma-Aldrich, St. Louis, MO, USA), supplemented with 10% fetal bovine serum (FBS) (Gibco, Grand Island, NY, USA), 100 U penicillin, and 100 μg/mL streptomycin (Gibco, Grand Island, NY, USA) and cultured in an atmosphere containing 5% CO_2_ at 37 °C. The medium was renewed two times a week. As described elsewhere [8,23]. we cultured rat INS-1 insulinoma cells with DOX-inducible expression of wild-type mouse IRE1α in RPMI-1640, 10% FBS, 1-mM sodium pyruvate, 10-mM HEPES, 2-mM glutamine, and 50-μM β-mercaptoethanol; these cells were also grown in 5% CO_2_ at 37 °C and passed by trypsinization when they reached >80% confluence. The culture medium was changed every 2–3 days.

### 4.3. Reagents

KIRA8 was house-made as previously reported [10]. Bortezomib (CAS#179324-69-7) was purchased from AdooQ BioScience (Irvine, CA, USA), thapsigargin (CAS#67526-95-8) from Sigma-Aldrich (St. Louis, MO, USA), and nilotinib (CAS#641571-10-0) from MedChemExpress (Monmouth, NJ, USA). Two PERK inhibitors, GSK2606414 (CAS#1337531-89-1) and AMG PERK 44 (AMG; CAS# 1883548-84-2), and dominant Polo-like kinase 2 (PLK2) inhibitor TC-S 7005 (CAS#1082739-92-1) were purchased from TOCRIS Bioscience (Bristol, UK). All reagents were dissolved in dimethyl sulfoxide (DMSO) to a final concentration of 10 mM as a stock solution and stored at −30 °C.

### 4.4. Cell Viability Assay

The CCK-8 from Dojindo (Tokyo, Japan) was used to measure the cell viability. IM-9 cells were seeded into a 96-well plate, with a density of 8 × 10^3^ cells/well. Then, the cells were treated as indicated with each drug for 24 or 72 h. Briefly, we added 10 μL CCK-8 solution to each well, and the plates were incubated for an additional 2 h at 37 °C. The optical densities at 450 and 650 nm were measured in a Corona plate reader SH-9000 (Hitachi, Tokyo, Japan). Then, the cell viability of the IM-9 cells was further confirmed using the trypan blue dye exclusion assay. The cells (2 × 10^5^ cells/mL) were inoculated into a 12-well plate and administered vehicle (DMSO) or KIRA8 (0.1 to 10 μM) for 60 min (min) followed by bortezomib (5 nM or 10 nM) or thapsigargin (500 nM) for 24 h, respectively. Next, the cells were stained with 0.4% trypan blue solution (Gibco, Grand Island, NY, USA), and the viable cells were counted using a hemocytometer counting chamber (Burker-turk, Erma Inc, Tokyo, Japan).

### 4.5. Apoptosis Assay by Flow Cytometry

We determined the apoptosis of myeloma cells using the annexin V–FITC Apoptosis Detection Kit (APOAF; Sigma-Aldrich, St. Louis, MO, USA) per the manufacturer’s instructions. Briefly, the cells were collected and resuspended in a binding buffer after washing with phosphate-buffered saline, and 0.5 μL fluorochrome-conjugated annexin V and 1 μL propidium iodide (PI) solution was added into 100 μL of cell suspension to incubate for 20 min at room temperature (RT) in the dark. Then, we calculated the percentage of apoptotic cells by flow cytometry analysis using FACSCalibur™ and FACSVerse™ Flow Cytometer (BD Biosciences, San Jose, CA, USA). The data analysis was performed using the FlowJo software v10.4.2 (Tree Star, Ashland, OR, USA). We defined the annexin-V positive/PI negative cells as cells in apoptosis.

### 4.6. Extraction of the Total RNA, Quantitative Real-Time Polymerase Chain Reaction, and the Detection of XBP1 Splicing

The total RNA of human myeloma cell lines and BM samples isolated by Ficoll solution (Histopaque-1077^®^; Sigma-Aldrich, St. Louis, MO, USA) was extracted using the RNeasy Mini Kit (Qiagen, Hilden, Germany) and reverse-transcribed to complementary DNA (cDNA) using the Omniscript Reverse Transcription Kit (Qiagen, Hilden, Germany). The samples were stored at −80 °C. We evaluated the purity and quality of the total RNA spectrophotometrically using NanoDrop One (Thermo Fisher Scientific, Tokyo, Japan). In addition, the quantitative RT-PCR was performed using the PowerUp SYBR Green Master Mix (Applied Biosystems, Foster City, CA, USA). All quantitative calculations were performed using the ΔΔCt methods. Appendix A lists the nucleotide sequences of primers used for PCR [36]. The gene expression levels were normalized with the expression of the housekeeping gene, *ACTB*. The amplification reactions were initiated by incubation at 95 °C for 5 min, followed by a three-step amplification at 95 °C for 15 s (s), 60 °C for 15 s, and 72 °C for 60 s for 40 cycles. All reactions were performed in triplicate. The *XBP1* mRNA processing was measured by amplifying the XBP1 cDNA with the primers: *XBP1* 5′-AAACAGAGTAGCAGCTCAGACTGC-3′ and 5′-GGATCTCTAAAACTAGAGGCTTGGTG-3′. The PCR products of *XBP1* cDNA were digested with *Pst*I, resolved on a 3% agarose gel, and stained with ethidium bromide [10,23].

### 4.7. RNA-Seq and Gene Expression Analysis

For the RNA-seq analysis, we isolated the total RNA as described previously. The samples were collected in duplicates from IM-9 cells treated with DMSO (as a control), 10-μM KIRA8 alone, and its combination treatment with 5 nM bortezomib or 500 nM thapsigargin. Using an Agilent Bioanalyzer 2100 and an RNA 6000 Nano Kit (Agilent Technologies, Santa Clara, CA, USA), the RNA integrity number (RIN) and concentration were determined. Then, 500 ng of total RNA (RIN > 8) of each sample was used for the cDNA library construction using a TruSeq Stranded mRNA Library Kit (Illumina Inc., San Diego, CA, USA) per the supplier’s protocol. In addition, the paired-end sequencing of cDNA libraries was obtained using the Illumina HiSeq 2500 sequencing platform with a 125-bp read length. The gene expression analysis, including hierarchical clustering and top 30 most downregulated genes, was performed by Genble Inc. (Fukuoka, Japan). Next, the image output data from the sequencer were transformed into raw sequence data by base calling and stored in the FASTQ format. 

Quality-controlled FASTQ sequences were trimmed using Prinseq-lite and Trimmomatic to remove poly (A/T) tails and low-quality reads, followed by mapping to the reference human genome GRCh38.p12 by STAR and measuring the transcript expression level in transcripts per million (TPM) with RNA-seq by expectation-maximization. Briefly, the samples were first grouped to compared with the pairwise control-treatment groups. Next, hierarchical clustering was calculated using the furthest neighbor method, with an averaged transcript expression level (TPM > 3, *n* = 2), and the results were displayed in the heatmap. The differentially expressed genes (DEGs) were screened per the following default criteria: fold change ≥2 and ≤0.5. Appendix A shows the raw data.

### 4.8. Cell-Cycle Analysis

IM-9 cells were centrifuged at 230× *g* for 5 min, and the supernatants were removed. Whilst vortexing, we gradually added 1 mL of ice-cold 70% ethanol and then fixed overnight at 4 °C. The fixed cells were centrifuged at 1000× *g* for 10 min, and the residual ethanol in the conical tube was left. Each sample was incubated in 1.0 mL of staining solution containing 1 mg/mL RNase and 50-µg/mL propidium iodide (Sigma-Aldrich, St. Louis, MO, USA) for >30 min at RT in the dark, and finally analyzed on the FACSCalibur™ and FACSVerse™ Flow Cytometer (BD Biosciences, San Jose, CA, USA). We calculated the percentage of cells in each phase of the cell cycle using the following method. At the peak of DNA content, 2N was defined as the G_0_/G_1_ phase, 4N as the G_2_/M phase, and the area between both peaks was defined as the S phase. Then, the respective cell cycle ratios were calculated by twice the area smaller than 2N as the G_0_/G_1_ phase, and twice the area larger than 4N as the G_2_/M phase, and the remaining proportion as the S phase.

### 4.9. Western Blotting

Briefly, human myeloma cells were lysed completely in Laemmli Sample Buffer with 2-mercaptoethanol (196-11022; Wako, Osaka, Japan) and were heated in a boiling water bath for 5 min. The samples were separated in sodium dodecyl sulphate–polyacrylamide gel electrophoresis (SDS–PAGE) and transferred to polyvinylidene fluoride membranes. To detect the PLK2 protein, the blots were incubated overnight at 4 °C with rabbit anti-human PLK2 polyclonal antibody (Thermo Fisher Scientific, Tokyo, Japan; Cat# PA5-14094, RRID:AB_2167742, 1:200 dilution) or rabbit anti-human PLK2 monoclonal antibody (Cell Signaling Technology, Berverly, MA, USA; Cat# 14812, RRID:AB_2798626, 1:500 dilution) for the samples obtained from INS1 cells. Then, the blots were incubated with horseradish peroxidase (HRP)–conjugated goat anti-rabbit IgG polyclonal antibody (Abcam, Toronto, ON, Canada; Cat# ab6721, RRID: AB_955447, 1:1000 dilution) at RT for 1 h. To detect β-actin, we used mouse anti-human β-actin monoclonal antibody (Sigma-Aldrich, St. Louis, MO, USA; Cat# A2228, RRID: AB_476697, 1:1000 dilution) and HRP-conjugated rabbit anti-mouse IgG antibody (Innovative Research, Novi, MI, USA; Cat# 61-6020, RRID:AB_88249, 1:1000 dilution) as primary and secondary antibodies, respectively. The protein bands were visualized by SuperSignal^®^ West Dura Substrate (Thermo Fisher Scientific, Tokyo, Japan) and exposed to X-ray film (FUJIFILM Medical Systems, Tokyo, Japan). We measured the band intensities using Image J software version 1.51k (National Institutes of Health, Bethesda, MD, USA). Notably, the β-actin level was used as an internal control to ensure equal amounts of loading proteins.

### 4.10. Immunohistochemical Analysis

We performed immunohistochemical analysis on formalin-fixed BM clots obtained from patients with NDMM [37,38]. In addition, we performed antigen retrieval using a microwave and a citrate buffer (pH 6.0). Then, tissue samples were incubated at RT for 60 min with an antibody specific for PLK2 (Thermo Fisher Scientific, Tokyo, Japan, Cat# PA5-14094, RRID: AB_2167742, 1:100 dilution). Next, DAB (3,3′-diaminobenzidine) staining was performed using the two-step EnVision+ System-HRP methodology (Dako, Tokyo, Japan) and reacted with hematoxylin. For double-label fluorescent immunohistochemistry, the samples were reacted with Cy2-conjugated goat anti-rabbit IgG polyclonal antibody (Jackson ImmunoResearch Labs, West Grove, PA, USA; Cat# 111-225-144, RRID: AB_2338021, 1:400 dilution) and PE-labelled mouse anti-human CD138 monoclonal antibody (BioLegend, San Diego, CA, USA; Cat# 352306, RRID: AB_10901158, 1:10 dilution). Finally, we used DAPI (4′,6-diamidino-2-phenylindole) as nuclear staining. Light and fluorescent images were captured using a BIOREVO BZ-X800 fluorescence microscope (Keyence, Tokyo, Japan).

### 4.11. Statistical Analysis

All experiments in this study were independently repeated, at least three times. The sample number is indicated in figures and figure legends. All values were expressed as the mean ± standard error of the mean (SEM). All statistical analyses were performed using GraphPad Prism version 6.00 (GraphPad Software Inc., San Diego, CA, USA) and JMP^®^ Pro 14 (SAS Institute Inc., Cary, NC, USA). In addition, Student’s *t*-test or one-way analysis of variance, followed by post-hoc Tukey’s test, were used to assess the statistical difference between two groups or between more than two groups, respectively, unless otherwise noted. Statistical significance was considered at *p* < 0.05. 

## Figures and Tables

**Figure 1 ijms-21-06314-f001:**
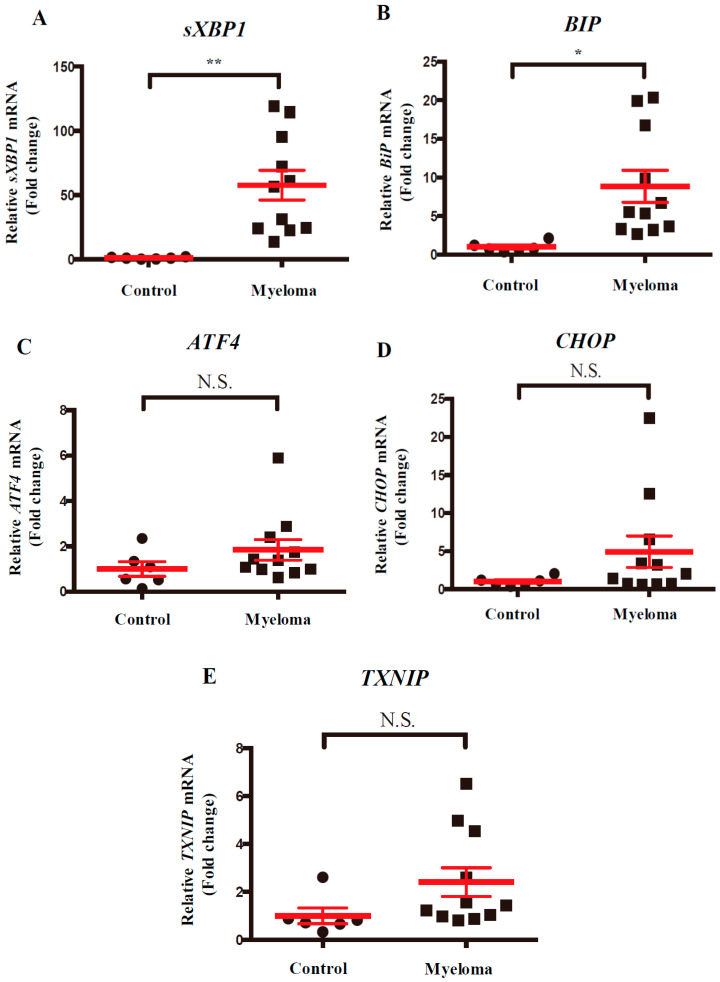
Unfolded protein response (UPR) markers in the bone marrow (BM) of patients with newly diagnosed multiple myeloma (NDMM). Quantitative real-time polymerase chain reaction (RT-PCR) of the relative mRNA levels of *sXBP1* (**A**), *BIP* (**B**), *ATF4* (**C**), *CHOP* (**D**), and *TXNIP* (**E**) in BM samples of patients with NDMM (*n* = 11) and control subjects (*n* = 6). Each symbol denotes an individual patient. The data shown are presented as the mean ± standard error of the mean (SEM). N.S., non-significant. * *p* < 0.05, ** *p* < 0.01.

**Figure 2 ijms-21-06314-f002:**
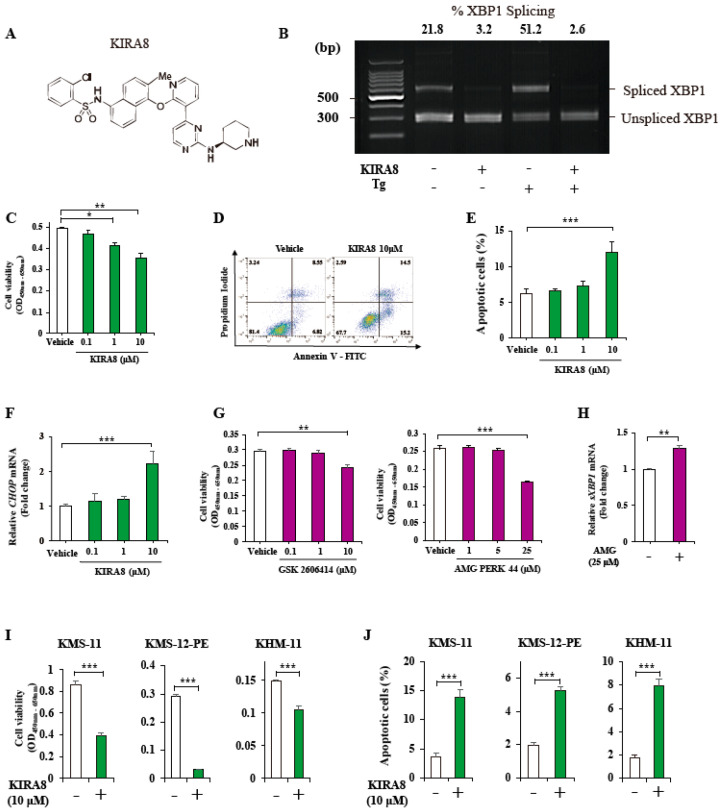
Effects of kinase-inhibiting RNase attenuator 8 (KIRA8) and protein kinase R-like endoplasmic reticulum kinase (PERK) inhibitors on human myeloma cells. (**A**) The structure of KIRA8 (**B**) *Pst* I-digested X-box Binding Protein 1 (XBP1) cDNA amplicons from IM-9 cells treated for 24 h (h) with vehicle (dimethyl sulfoxide, DMSO) or 10 μM of KIRA8 and 500 nM of thapsigargin (Tg). Top, the intensity ratios of the spliced form to the total *XBP1*. (**C**–**E**) IM-9 cells were treated with vehicle (DMSO) or KIRA8 with the indicated concentrations for 24 h. (**C**) The cell viability was assessed using the cell counting kit-8 (CCK-8) assay. (**D**,**E**) Apoptosis was assessed using FACS analysis of annexin V/propidium iodide (PI)–stained myeloma cells treated with vehicle or KIRA8 for 24 h. Annexin V-positive/PI-negative cells are regarded as cells in apoptosis. Representative FACS plots show annexin V/PI-stained cells (**D**) and the % apoptotic cells (**E**). (**F**) Quantitative RT-PCR of the relative *CHOP* mRNAs from IM-9 cells treated with vehicle (DMSO) or KIRA8 for 24 h. (**G**) The cell viability in IM-9 cells treated with GSK 2606414 or AMG for 24 h. (**H**) Quantitative RT-PCR of the relative *sXBP1* mRNA levels from IM-9 cells treated with vehicle (DMSO) or 25 μM of AMG for 24 h. (**I**,**J**) The cell viability (**I**) and apoptosis (**J**) assay in three different human myeloma cells—KMS-11, KMS-12-PE, and KHM-11—treated with vehicle (DMSO) or 10 μM of KIRA8 for 24 h. The data shown are the mean ± SEM. For all experiments, three independent biological samples were used. *, *p* < 0.05; **, *p* < 0.01; ***, *p* < 0.001.

**Figure 3 ijms-21-06314-f003:**
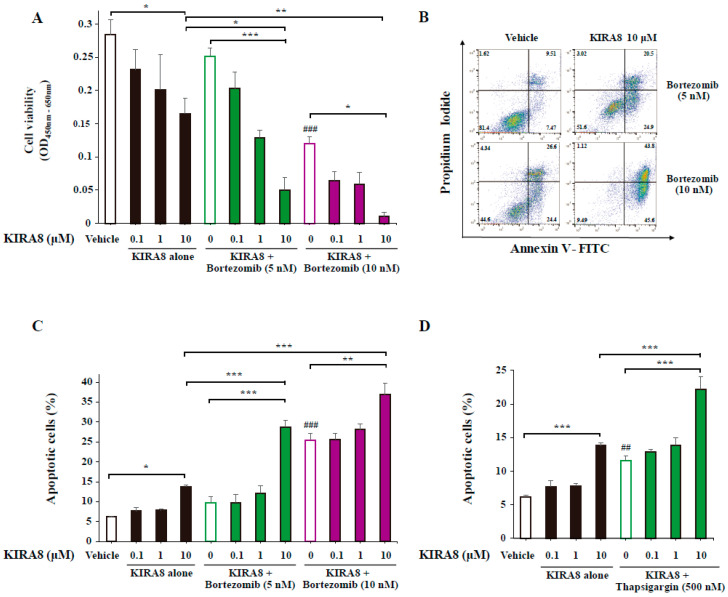
Anti-tumor effects from a combination with KIRA8 and bortezomib in IM-9 cells. (**A**) The cell viability assessed by the CCK-8 assay in IM-9 cells treated with vehicle (DMSO) or the indicated concentrations of KIRA8 for 1 h, followed by the indicated concentrations of bortezomib for 24 h. Three independent biological samples were used. (**B**,**C**) The % apoptotic cells in IM-9 cells treated with vehicle (DMSO) or the indicated concentrations of KIRA8 for 1 h, followed by the indicated concentrations of bortezomib for 24 h. The representative FACS plots show annexin V/PI–stained cells (**B**) and the % apoptotic cells (**C**). Four independent biological samples were used. (**D**) The % apoptotic cells in IM-9 cells treated with vehicle (DMSO) or the indicated concentrations of KIRA8 for 1 h, followed by 500 nM of Tg for 24 h. Four independent biological samples were used. The data shown are the mean ± SEM. *, *p* < 0.05; **, *p* < 0.01; ***, *p* < 0.001. ^##^, *p* < 0.01; ^###^, *p* < 0.001; vs. vehicle.

**Figure 4 ijms-21-06314-f004:**
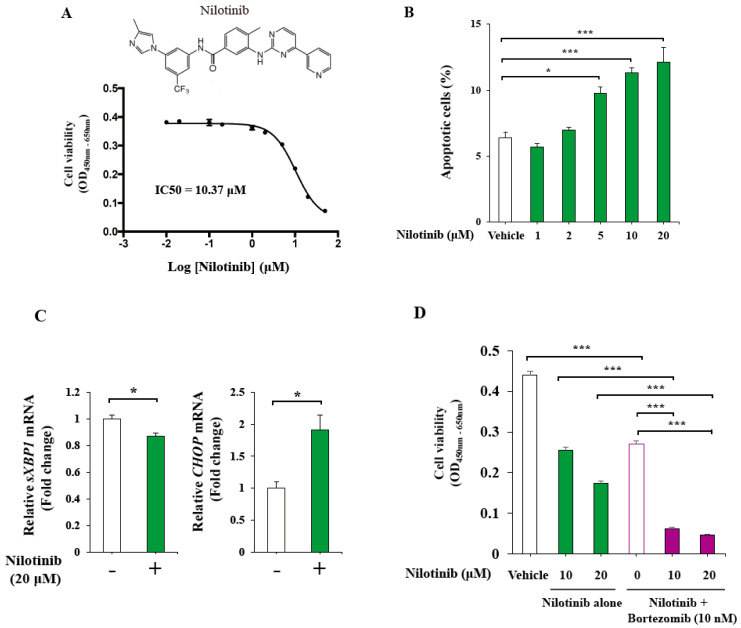
Anti-myeloma effects of nilotinib in IM-9 cells. (**A**) The structure of nilotinib and the cell viability assessed by the CCK-8 assay in IM-9 cells treated with the indicated concentrations of nilotinib for 24 h (*n* = 4). (**B**) The % apoptotic cells in IM-9 cells treated with vehicle (DMSO) or the indicated concentrations of nilotinib for 24 h (*n* = 3). (**C**) Quantitative RT-PCR of the relative *sXBP1* and *CHOP* mRNA levels from IM-9 cells treated with vehicle (DMSO) or 20 μM of nilotinib for 24 h (*n* = 3). (**D**) The cell viability assessed by the CCK-8 assay in IM-9 cells treated with vehicle (DMSO) or the indicated concentrations of nilotinib and bortezomib for 24 h (*n* = 4). The data shown are as the mean ± SEM. *, *p* < 0.05; **, *p* < 0.01; ***, *p* < 0.001.

**Figure 5 ijms-21-06314-f005:**
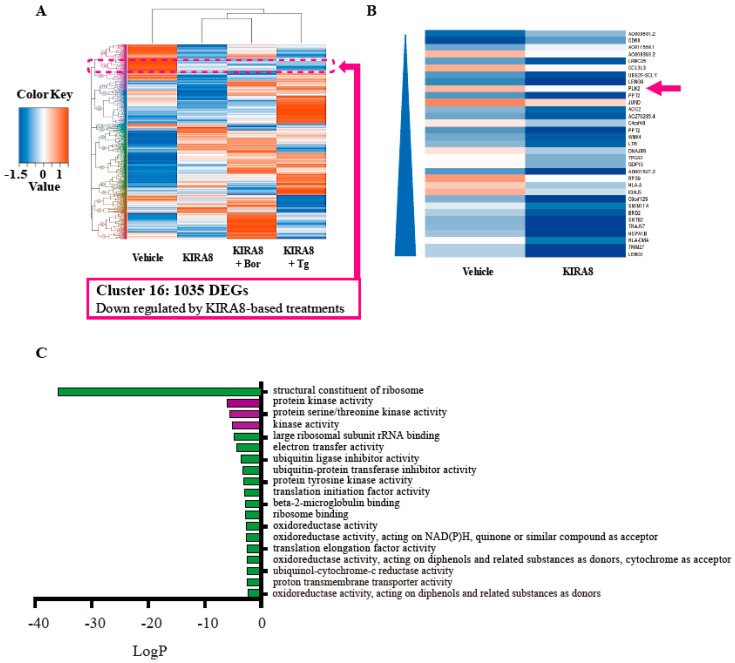
The gene expression profile by KIRA8 in human myeloma cells. (**A**) IM-9 cells were treated with vehicle (DMSO), 10 μM of KIRA8, and its combination with 5 nM of bortezomib (Bor) or 500 nM of Tg in duplicate. The heatmap shows the cluster analysis from each group. Cluster 16 (dotted frame) represents the differentially expressed genes (DEGs) suppressed by KIRA8-based treatments. (**B**) The gene list shows the top 30 downregulated protein-coding DEGs treated with KIRA8 alone; the most downregulated gene is at the bottom. Pink arrow, the *Polo-like kinase 2* (*PLK2*) was focused on in this study. (**C**) The list of Gene Ontology-enriched terms for molecular function in cluster 16. Terms related to PLK2 are highlighted in purple. (**D**) Quantitative RT-PCR of the relative *PLK1* and *PLK2* mRNA levels from IM-9 cells treated with vehicle (DMSO) or 10 μM of KIRA8 for 24 h. For these experiments, three independent biological samples were used. (**E**) Western blotting of protein extracts from IM-9 cells treated with vehicle (DMSO) or 10 μM of KIRA8 for 24 h using anti-PLK2 antibodies. Numbers above the representative western blotting picture denote the signal intensity ratios of PLK2 and β-actin measured by Image J software. The β-actin level was used as an internal control. For statistical analysis, three independent biological samples data were used. The data shown are the mean ± SEM. *, *p* < 0.05; **, *p* < 0.01.

**Figure 6 ijms-21-06314-f006:**
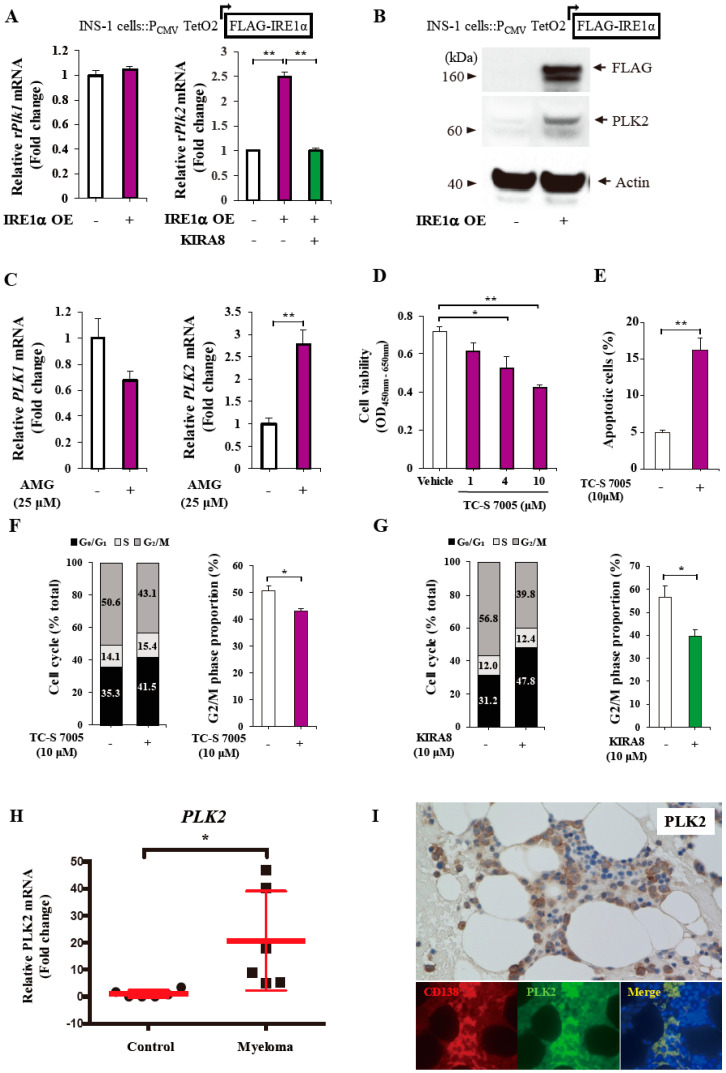
PLK2 expression was IRE1α-dependent and its inhibitor exhibited anti-myeloma effects in IM-9 cells. PLK2 protein and mRNA were expressed strongly in patients with NDMM. (**A**,**B**) Quantitative RT-PCR of relative rat *Plk1* and *Plk2* mRNA levels in duplicate (**A**) and western blotting of PLK2 protein (**B**) from doxycycline (DOX)-inducible IRE1α-overexpressing INS-1 insulinoma cells treated with vehicle (DMSO) or 0.5 μM of KIRA8 with or without 1 μg/mL of DOX for 24 h. (**C**) Quantitative RT-PCR of the relative *PLK1* and *PLK2* mRNA levels from IM-9 cells treated with 25 μM of AMG for 24 h (*n* = 3). (**D**) The cell viability in IM-9 cells treated with vehicle (DMSO) or the indicated concentrations of TC-S 7005 for 72 h (*n* = 3). (**E**) Annexin V-positive/PI-negative IM-9 cells treated with vehicle (DMSO) or 10 μM of TC-S 7005 for 72 h (*n* = 3). (**F**,**G**) The % components of cell-cycle phases and % cells in the G_2_/M phase in IM-9 cells treated with vehicle (DMSO) and 10 μM of TC-S 7005 for 72 h (**F**) or 10 μM of KIRA8 (**G**) for 24 h. For these experiments, four independent biological samples were used. (**H**) Quantitative RT-PCR of relative *PLK2* mRNA levels in BM samples of control subjects (*n* = 6) and patients with NDMM who attained a complete response receiving bortezomib-based treatment and high-dose melphalan (*n* = 6). (**I**) The expression of PLK2 protein in the BM clot specimen of a patient with NDDM was confirmed by immunohistochemical analysis with anti-PLK2 antibodies (magnification, ×200). The rabbit immunoglobulin G control was negative (data not shown). Immunofluorescence at higher magnification illustrates the expression of the PLK2 protein (green) in CD138^+^ myeloma cells (red). DAPI (4′,6-diamidino-2-phenylindole; blue) was used as a nuclear counterstain. The data shown are the mean ± SEM. *, *p* < 0.05; **, *p* < 0.01.

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
