# Peer review of "Targeting Adaptive IRE1α Signaling and PLK2 in Multiple Myeloma: Possible Anti-Tumor Mechanisms of KIRA8 and Nilotinib"

_ijms, 2020, doi:10.3390/ijms21176314_

Round 1

Reviewer 1 Report

The manuscript by Yamashita et al describes the effect of KIRA8 and Nilotinib on IRE1 in myeloma cells, looking at various aspects including expression of XBP1, CHOP, PLK2, cell death, combined effects with bortezomib and thapsigargin. The manuscript contains some interesting observations, particularly the high levels of PLK2 in myeloma patient samples (albeit with low sample numbers). However, the manuscript does not have a clear focus and in my opinion some of the conclusions are not warranted, specifically regarding induction of apoptosis as the mode of cell death, and induction of synergy. Specific comments are given below:

Results

Section 2.1 - The relevance of measuring ATF6 mRNA as a measure of UPR is not clear to me, since it is mainly regulated during UPR by proteolysis of the precursor protein

Fig 2B – usually the spliced form of xbp1 has a faster mobility than the unspliced form. Can the authors explain this?

In fig 2E and 2J the authors have only examined annexin V staining, and not whether the cells are also negative for PI. So while there is an increase in cell death, it is not true to claim that there is an increase in apoptosis. (However, in Fig 2D I agree that there is an increase in apoptosis – the lower right quadrant of the graph, as well as an increase in necrosis – the upper right quadrant).

Section 2.3 and Fig 3 – the authors say that there is an increase in apoptosis. However, though I agree that there is an increase in cell death, the cells are stained positive for PI which means that they are necrotic not apoptotic. Apoptotic cells are those that stain positive for Annexin V and negative for PI.

Same comment for fig 4B,

Fig 3A-D – the control is missing for KIRA8 alone in the absence of bortezomib or thapsigargin. Also this figure does not show synergy. A combination index should be determined in order to comment on synergy.

Fig 4D – the control is missing for nilotinib alone in the absence of bortezomib. Also this figure does not show synergy – in fact, looking at figure 4A nilotinib alone seems to be toxic at the doses being used in fig 4D. However, these controls should be done in the same experiment

The flow of the logic is very disjointed. The rationale for using nilotinib is not clear, what is the point of using a non-specific inhibitor of IRE1 such as nilotinib? What is the purpose of examining the effect of nilotinib on xbp1 and chop levels

Fig 5A – the key in the top left is too small to read. I have to assume that red means high expression and blue means low.

Controls are missing from the RNAseq experiment, ie bortezomib alone and thapsigargin alone

Line 252 – ‘under the induction of sXB1’ there is only correlative evidence for this. The increase in PLK2 could also be due to inhibition of RIDD

Line 279-280 – the term ‘attained a stringent complete response’ is not clear

Discussion

Line 317 - The authors state ‘the detailed mechanisms of how KIRA8 works remained unclear’. However this manuscript does not contribute towards understanding the detailed mechanisms. There is no examination of effect on other aspects of IRE1 biology such as RIDD, interaction with other proteins, IRE1 oligomerisation, etc.

Line 318 – the authors claim that there is ‘the reciprocal activation of PERK’. However this is not substantiated by the data presented. There are several alternative explanations for the results they present, including ISR activation

Line 337-338 – the authors describe PLK2 as a ‘key downstream target of the IRE1–XBP1 pathway’. The data do not show that PLK2 is a downstream target of IRE1-XBP1. It could just as easily be a RIDD target and not an XBP1 target.

The induction of apoptosis is not clear from the data.

The experiments to demonstrate synergy lack key controls.

General: Language needs attention for English grammar and formatting.

Author Response

Dear Reviewer #1,

Thank you for your detailed reading and suggestions on our manuscript. We are pleased that you appreciated the major instructive results of the work and recommended publication of a revised version. Your suggestions and comments were very helpful, and during the revision phase we have made every attempt to address your questions by further analyzation of the data or through clarifications in the narrative (please see point-by-point responses, below). Through addressing your suggestions, the manuscript is now much improved. With these changes, we hope that the manuscript is now acceptable to you for publication.

Best wishes,

Yamashita, et. al.

Specific comments are given below:

Results

  1. Section 2.1 - The relevance of measuring ATF6 mRNA as a measure of UPR is not clear to me, since it is mainly regulated during UPR by proteolysis of the precursor protein

Response: As suggested, we have deleted all the information regarding ATF6 mRNA from the manuscript, Fig 1C amd Table S2.

  1. Fig 2B – usually the spliced form of xbp1 has a faster mobility than the unspliced form. Can the authors explain this?

Response: We respectfully explain that, in this assay, we have digested amplicon of unspliced XBP-1 mRNA with specific enzyme PstI, thus, we can observe two “digested” bands around 300bp. This method could separate pure unspliced bands from spliced/unspliced XBP1 hybrid amplicon, which resulted in accurate calculation of % spliced/unspliced XBP1 mRNA. We have previously presented the detail of this methods (Han D, Papa FR, et al, Cell2009;138:562-575). To clarify, we have cited this paper in the Materials and Methods section 4.6.

  1. In fig 2E and 2J the authors have only examined annexin V staining, and not whether the cells are also negative for PI. So while there is an increase in cell death, it is not true to claim that there is an increase in apoptosis. (However, in Fig 2D I agree that there is an increase in apoptosis – the lower right quadrant of the graph, as well as an increase in necrosis – the upper right quadrant).

Response: As suggested, we have reanalyzed the original data and presented the Annexin V-positive and PI-negative apoptotic cells (only in the lower right quadrant of the graph) as the cells in apoptosis in Fig 2E & J. In addition, we added the information about the apoptosis in Material and methods section (4.5) and Figure legends (Figure 2D and E) in the revised version.

  1. Section 2.3 and Fig 3 – the authors say that there is an increase in apoptosis. However, though I agree that there is an increase in cell death, the cells are stained positive for PI which means that they are necrotic not apoptotic. Apoptotic cells are those that stain positive for Annexin V and negative for PI.

Response: As mentioned in #3, we have defined Annexin V-positive and PI-negative cells as apoptotic cells and presented the results in Fig 3 and have modified this legend.

  1. Same comment for fig 4B,

Response: Same as #3 and #4, we have also reanalyzed the original data and modified Fig4B and related parts in Figure legends of Fig4.

  1. Fig 3A-D – the control is missing for KIRA8 alone in the absence of bortezomib or thapsigargin. Also this figure does not show synergy. A combination index should be determined in order to comment on synergy.

Response: As suggested, we have added the control data treated with KIRA8 alone in Fig 3A, 3C and 3D. We agree this point that the data we showed was not enough to provide the evidence of synergetic effects. We have modified the description to be accurate the related part in Result section 2.3 in revised manuscript.

  1. Fig 4D – the control is missing for nilotinib alone in the absence of bortezomib. Also this figure does not show synergy – in fact, looking at figure 4A nilotinib alone seems to be toxic at the doses being used in fig 4D. However, these controls should be done in the same experiment

Response: We agree that the original figure 4D was confusable. To clarify that there are the controls with Nilotinib alone, we have added the tag “Nilotinib alone” in X-axis of Fig 4D to avoid confusion. Also, we modified the related parts regarding the synergy effects.

  1. The flow of the logic is very disjointed. The rationale for using nilotinib is not clear, what is the point of using a non-specific inhibitor of IRE1 such as nilotinib? What is the purpose of examining the effect of nilotinib on xbp1 and chop levels

Response: Thank you for highlighting the important point. Here is our rationale. We totally agree that nilotinib is non-specific inhibitor of IRE1. However, from the view point of clinical aspect, we believe that it is important to demonstrate the anti-myeloma effect of FDA-approved drugs which inhibit IRE1α activation. We respectfully selected nilotinib with these two reasons: First, among the FDA-approved drugs, high-throughput screening using a DELFIA assay identified nilotinib as a promising candidate of IRE1α autophosphorylation (Newbatt Y, et al. J Biomol Screen 2013;18:298-308). Further, we have proved that nilotinib exerted the robust inhibition of IRE1a activity with the IC50 of 3.21 mM for attenuating XBP1 splicing (Morita S, Papa FR, et al, Cell Metab, 2017;25:883-897). The purpose of examining the effect of nilotinib on xbp1 and chop levels are showing the anti-IRE1a effects and suggestive reciprocal activation of PERK, as KIRA8. To clarify, we added the explanation of our logic in Result section (Result section 2.4).

  1. Fig 5A – the key in the top left is too small to read. I have to assume that red means high expression and blue means low.

Response: As suggested, we have enlarged the key in the top left corner of the Fig 5A.

  1. Controls are missing from the RNAseq experiment, ie bortezomib alone and thapsigargin alone

Response: As pointed out, we agree that there are some limitations in our cluster analysis of RNA-seq due to miss the controls of KIRA-based treatments. We clarified the limitation of cluster analysis in RNA-seq results to interpret the findings carefully for the readers in third to sixth lines in fourth paragraph in the Discussion section.

  1. Line 252 – ‘under the induction of sXB1’ there is only correlative evidence for this. The increase in PLK2 could also be due to inhibition of RIDD

Response: We agree this point. We have reworded this sentence as follows: “AMG increased the mRNA expression of PLK2, but not PLK1, potentially under the induction of sXBP1 or inhibition of regulated IRE1-dependent decay (RIDD)” in last sentence in first paragraph in Result section 2.6 in the revised version.

  1. Line 279-280 – the term ‘attained a stringent complete response’ is not clear

Response: This term had meant complete response defining negative immunofixation and normal bone marrow in MM patients. To make it more easily to understand, we have deleted the phrase “stringent” in the related part in the section 2.7. and Figure 6.

Discussion

  1. Line 317 - The authors state ‘the detailed mechanisms of how KIRA8 works remained unclear’. However this manuscript does not contribute towards understanding the detailed mechanisms. There is no examination of effect on other aspects of IRE1 biology such as RIDD, interaction with other proteins, IRE1 oligomerisation, etc.

Response: As you pointed out, we expressed this part in Discussion section more carefully to avoid misunderstanding. We modified them in fourth to eighth lines in the second paragraph of Discussion section.

  1. Line 318 – the authors claim that there is ‘the reciprocal activation of PERK’. However this is not substantiated by the data presented. There are several alternative explanations for the results they present, including ISR activation

Response: We agree that there could be several explanations other than PERK activation for the results you mentioned. We have modified the related part in third and last paragraphs in ABSTRACT and second and last paragraphs in Discussion section in the main text.

  1. Line 337-338 – the authors describe PLK2 as a ‘key downstream target of the IRE1–XBP1 pathway’. The data do not show that PLK2 is a downstream target of IRE1-XBP1. It could just as easily be a RIDD target and not an XBP1 target.

Response: We thank you for highlighting this important point. We agree that PLK2 could indirectly be a RIDD target through decay of the target mRNAs which stabilize PLK2 such as microRNAs, or induce PLK2 expression. We modified the related part in the first to third lines in the fourth paragraph in the Discussion section.

  1. The induction of apoptosis is not clear from the data.

Response: As you suggested, we defined Annexin V-positive/PI-negative cells as the cells in apoptosis as we responded in comments #3-5 related to the Result section. We clarified the definition and reanalysis all the results obtained from FACS analysis. We modified the Figures 2E, 2J, 3C, 3D, 4B and 6E and the related these legends and in fourth to fifth and ninth lines in Methods section 4.5 in main text.

  1. The experiments to demonstrate synergy lack key controls.

Response: As we responded in #6, we agree that our data is not enough to show the synergy effects clearly. Therefore, we have modified the expression in the related part in Result section 2.3 in revised manuscript to avoid the confusion.

  1. General:Language needs attention for English grammar and formatting.

Response: As you suggested, this manuscript has been proofread by a MDPI English Editing Services in this revision phase.

Reviewer 2 Report

The study is well performed, clearly presented and informative.The data warrant the conclusions. 

There are are only two minor corrections needed:

Line 109: (Figure B) should read Figure 2B

Line 197: hypnotized should read hypothesized

Author Response

Dear Reviewer #2,

Thank you for your detailed reading our manuscript. We are pleased that you found the article to be well performed, clearly presented and informative. During the revision phase we have correct the points you suggested. Through addressing your suggestions, the manuscript is now much improved. With these changes, we hope that the manuscript is now acceptable to you for publication.

Best wishes,

Yamashita, et. al.

There are are only two minor corrections needed:

1. Line 109: (Figure B) should read Figure 2B

Response: As kindly suggested, we have corrected the figure number.

2. Line 197: hypnotized should read hypothesized

Response: As suggested, we have corrected the typo.

Round 2

Reviewer 1 Report

The authors have made revisions that satisfy my original comments and suggestions.